# Progress in Research and Application of Graphene Aerogel—A Bibliometric Analysis

**DOI:** 10.3390/ma16010272

**Published:** 2022-12-27

**Authors:** Bowen Chai, Wanlin Zhang, Yuanyuan Liu, Shuang Zhu, Zhanjun Gu, Hao Zhang

**Affiliations:** 1CAS Key Laboratory for Biomedical Effects of Nanomaterials and Nanosafety, Institute of High Energy Physics, Beijing 100049, China; 2School of Science, China University of Geosciences, Beijing 100083, China; 3Aerospace Research Institute of Special Material and Processing Technology, Beijing 100074, China; 4CAS Center for Excellence in Nanoscience, National Center for Nanoscience and Technology of China, Chinese Academy of Sciences, Beijing 100190, China; 5Center of Materials Science and Optoelectronics Engineering, College of Materials Science and Optoelectronic Technology, University of Chinese Academy of Sciences, Beijing 100049, China

**Keywords:** graphene aerogel, graphene oxide, composites, nanostructure, fabrication, energy storage, electrode, environmental protection, adsorption

## Abstract

In recent years, graphene aerogel (GA) has been widely used as a 3D porous stable network structure material. In order to identify the main research direction of GA, we use the bibliometric method to analyze its hot research fields and applications from the Web of Science database. First, we collected all relevant literature and analyzed its bibliometrics of publication year, country, institution, etc., where we found that China and Chinese Academy of Sciences are the most productive country and institute, respectively. Then, the three hot fields of fabrication, energy storage, and environmental protection are identified and thoroughly discussed. Graphene aerogel composite electrodes have achieved very efficient storage capacity and charge/discharge stability, especially in the field of electrochemical energy storage. Finally, the current challenges and the future development trends are presented in the conclusion. This paper provides a new perspective to explore and promote the related development of GA.

## 1. Introduction

Graphene was successfully extracted from graphite using mechanical exfoliation by Geim and Novoselov in 2004 [1]. As the first 2D crystalline material composed of single atoms, it is widely used because of its excellent physical and mechanical properties as well as its outstanding electrical conductivity, thermal conductivity, and optical properties [2,3]. At the same time, three-dimensional graphene materials such as graphene aerogel (GA), where graphene sheets are interconnected with each other to retain the unique properties of graphene, has a stable structure that can effectively reduce the problem of greatly reduced physical and chemical properties caused by Π-Π stacking due to the poor dispersion of graphene [4]. Besides, GA is easier to prepare and more efficient in performance compared to carbon nanotube-based 3D architectures [5]. Initially, graphene or graphene oxide was used as precursors, which were formed into graphene hydrogels (GH) by conventional sol-gel molding [6]. Since the internal structure collapses and shrinks in volume under capillary pressure during the normal freeze-drying process thus leading to severely impaired performance, GA is usually formed by supercritical drying to maintain the stability of the internal space [7]. With the development of science and technology, template method and 3D printing are also used to prepare GA with smaller pore size, isotropic, and different types of ordered network structures [8,9,10,11]. Meanwhile, a lot of current research has been conducted to achieve the desired performance enhancement through GA compounding with other materials, heteroatom doping, etc. For example, GA is compounded with transition metal nanoparticles to enhance electrical conductivity and maintain stability, or the surface-active sites of GA are increased by doping with N atoms to improve adsorption properties, etc. [12,13]. In terms of functions, GA has excellent properties such as low density, high electrical conductivity, high thermal insulation, large specific surface area, good resilience, and strong adsorption ability, which can effectively be used in important files including resource shortage and environmental pollution. Its high electrical conductivity and stable three-dimensional structure has electrodes in electrochemical energy storage devices for energy applications [14,15,16,17], its porous structure and resilience properties enable their application as an adsorbent and solar photocatalytic carrier in environmental remediation [18,19,20], and its compressibility and sensitivity to gas and liquid for use as sensors [21,22], etc. The emergence of graphene aerogels has greatly facilitated the development and application of graphene-related materials, which can be effectively used in important situations such as resource shortage and environmental pollution. Although GA research has only been developed for more than a decade, it is already feasible to be prepared on an industrial scale and has become one of the popular 3D materials.

Due to the wide range of GA applications, reviews that summarizes its application have been widely reported [23,24,25,26,27]. However, there is still a lack of a work that can provide an overview of the popular research areas from a holistic and objective perspective, which could help researchers to better understand and grasp the key application directions related to GA. Therefore, an objective and comprehensive statistical approach to GA-related research is needed. Based on this, we for the first time use bibliometric analysis in conjunction with the Web of Science repository to summarize and analyze the current published articles and results of GA. We first discuss the current status of research in the field of GA and the extent to which different countries/institutions have contributed to it. Then, according to the objectively obtained keyword mapping, the latest development trend of GA and the research focus including material preparation, energy storage application, and environmental protection application are identified and summarized. This work serves as a focus to show readers the current research hotspots in GA and the feasibility of future development in these three areas. Finally, we discuss the challenges encountered in the current development of GA, as well as the related views on its improvement and the outlook on the application in the future. Our work aims to help readers to grasp a general research status of GA research and further provide insights for guiding future development of this important field.

## 2. Bibliometric Analysis

### 2.1. Collection of Data

The data used for this review were retrieved from Web of Science on 4 October 2022. The preliminary results were obtained by advanced search “TS = Graphene Aerogel* not AK = (Foam* or Sponge*),” and then the keywords and abstracts of the retrieved literature were manually filtered to obtain the full WOS core collection of 5705 papers and 234 reviews.

It should be noted that the combination of the word Graphene aerogel first appeared in a paper published by Wang et al. in 2009 [28]. Since then, in many studies on carbon aerogels (CA), many commonalities can be reflected in GA, so they are also included.

### 2.2. Annual Publications and Countries/Institutes Contributing to GA research

After 13 years of research and development since 2009 (2009–2021), the number of published articles has grown to more than 900 in the last three years, as shown in Figure 1a. Besides, it is still in a steady growth trend, indicating that the field of GA is attracting more and more researchers’ attention. Among them, researchers from China contributed the most publications (4295/178,054 citations), followed by the United States (551/36,701 citations). There are also more than 200 publications were Korea (238 publications/8961 citations) and Australia (208 publications/12,672 citations). The criteria for judging contribution need to consider several indicators, such as publication number, citation number, and average citation number at the same time. From this point of view, China provides the highest number of articles while the United States has the highest average number of citations. Furthermore, according to the analysis using VOSviewer, cooperation ties exist between 48 different countries in Figure 1b. In the results of this analysis, China, as the country with the most cooperative relations, cooperates with 43 of them. The next largest country is the United States, which cooperates with 32 of these countries. The rest of the countries also have different collaborations with each other, indicating that extensive collaboration of each country. In addition, when it comes to institute publication, Chinese Academy of Sciences top the list in terms of the number of publications contributed (603) and h-index (96) in Figure 1c. Herein, the h-index reflects the number and level of academic output of researchers. For example, if somebody’s h-index is 10, it means that he has 10 articles, and each paper is cited at least 10 times. Lastly, the main disciplines covered in the WOS search are “Chemistry, Materials Science, Science Technology Other Topics, Physics.” This phenomenon indicates that the development of research on GA is a joint result of the intersection of multiple disciplines, and most of the research on GA is focused on the properties and applications of the material.

### 2.3. Research Hotspots

After common bibliometric analysis of GA research performance, we used VOSviewer to analyze the keywords of the articles to determine the hotspots in the GA research field, and selected keywords with more than 20 occurrences. We discarded some keywords like Graphene, Aerogel, Graphene aerogel, etc., which did not show any useful information in Figure 2. The larger the node, the more frequent it appears. The results presented in the figure show that the keywords “Fabrication, Supercapacitor, Adsorption, Performance” appear most frequently. Accordingly, the keywords are subjectively classified into three main research hotspots by Vosviewer. As shown in Figure 2, the green section contains the synthesis, structure, and performance of GA materials. The red section indicates the applications of GA in energy storage, mainly focusing on the applications of GA in electrochemical disciplines such as supercapacitors and electrode materials for battery systems. Finally, the blue section contains GA applications in the environmental protection field, including adsorption and removal of pollutants. In the next chapter, we will focus on each research hotspot of GA.

## 3. Fabrication of Graphene Aerogels

The fabrication of GAs are mainly classified according to the nodes presented in the green clusters. By employing different methods, diverse properties of GA could be excavated and obtained. At present, the main fabrication methods involved in the formation of graphene aerogels include self-assembly methods, template methods, and 3D printing methods.

### 3.1. Self-Assembly Method for GA Fabrication

#### 3.1.1. In-Situ Assembly Method

The in-situ assembly method is usually divided into two types: hydrothermal reductive assembly method and chemical reductive assembly method. Graphene oxide (GO) is used as a precursor, which is uniformly and stably dispersed in solution by the electrostatic repulsion of the surface negatively charged functional groups such as epoxy groups and carboxyl groups, which are shown in [29,30] Figure 3a. When the surface functional groups are removed by hydrothermal conditions or chemical reducing agents to eliminate the electrostatic repulsion, the GO layer sheets aggregate with each other. The graphene hydrogels are stacked into three-dimensional structures by the combined effects of van der Waals forces, dipoles, and hydrophilic/hydrophobic interactions, and finally freeze-dried to obtain GA [31].

Hydrothermal assembly utilizes high temperature and pressure conditions in closed conditions where graphene sheets in solution associate with each other to form a stable three-dimensional structure [7,30,32] Figure 3b,c. In this process, the pH value of the solution and the hydrothermal reaction time have a large influence on the structure [30]. First, the oxygen-containing groups on the GO surface are subject to different degrees of ionization by different pH values, which come to influence the final material properties. At low pH (pH < 3) levels, the surface of the synthesized hydrogel adsorbent is positively charged and has a strong adsorption effect on anions. On the contrary, when the pH increase leads to a decrease in the H^+^ concentration in the solution, the adsorbent is less protonated, causing a weakening of the adsorption capacity [33,34]. Second, the hydrothermal time affects the degree of GO reduction. Generally, the longer the hydrothermal time, the better the degree of reduction leads to the stronger performance of the formed graphene hydrogel. However, a reduction time that is too long leads to the increase in self-assembled cross-linked sites in the 3D structure and the increase in density due to the decrease in specific surface area and pore volume [35]. Therefore, ensuring a proper hydrothermal time in a low-pH environment can be expected to make the structure and properties of GA reach optimal values.

**Figure 3 materials-16-00272-f003:**
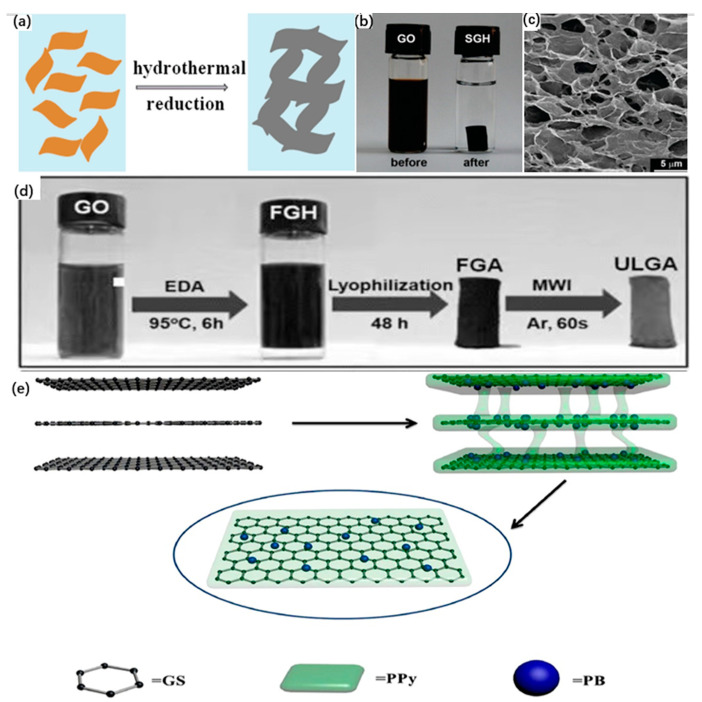
(**a**) The proposed formation mechanism for SGH; (**b**) Photos of GO aqueous dispersion after 12 h of hydrothermal assembly; (**c**) Photos of the internal microstructure of SGH under SEM [30]; (**d**) Illustration of the fabrication process of the ultralight graphene aerogel (ULGA) [7]; (**e**) Schematic diagram of polypyrrole (PPy) used as a crosslinking agent for GA assembly [36].

Compared to the harsh conditions of hydrothermal assembly, the synthesis of graphene hydrogels using chemical reductive assembly requires the addition of reducing agents (commonly used agents are sodium ascorbate, hydroquinone, ethylenediamine, etc.) to chemically reduce GO [31,37,38], which is more convenient to operate and thus widely adopted. Sheng et al. used sodium ascorbate to reduce GO to promote the three-dimensional assembly of graphene sheets, effectively removing the oxygen-containing functional groups between the carbon atom layers and providing a new environmentally friendly pathway for the reductive assembly of GA [31]. In addition, the addition of the reducing agent ethylenediamine (EDA) to the solution of GO resulted in the assembly of monolayers of functionalized graphene hydrogels. The EDA molecules bonded on at the same time reduced the stacking of the lamellae and there was almost no volume change before and after the treatment. the overall structure inside the GA was not disrupted and the integrity was preserved [39]. However, the addition of reducing agents introduces non-carbon impurities and some surface functional groups, so the amount of reducing agents needs to be strictly controlled or de-hybridized [38]. For example, Hu et al. used microwave irradiation to achieve the removal of surface impurities based on the reduction in graphene oxide solutions using EDA, thus reducing the adverse effects on the properties of 3D graphene materials [7], as shown in Figure 3d.

#### 3.1.2. Cross-Linker-Induced Self-Assembly Method

The prominent feature of cross-linker-induced self-assembly lies in the addition of chemical cross-linkers during the synthesis process, and the surfaces where GO and cross-linkers are bound form bond bridges could interlink into a three-dimensional network structure [36,40]. The graphene hydrogel sheets synthesized with the participation of crosslinkers have less interlayer resistance and stronger structure than the in-situ self-assembly method where physical linkage occurs [41]. Simultaneously, the physical and mechanical properties are better, and the electrical conductivity is enhanced by about two orders of magnitude [6]. Usually, the main metal ions (such as Ni^+^, Na^+^, Co^3+^, etc.) and organic compounds containing functional groups (mainly include acrylic acid, phenolic resin, and other polyacid esters) can be able to act as cross-linking agents to promote gel formation. Graphene aerogel materials with different properties can be achieved by adding different kinds of cross-linking agents. For example, Cong et al. reported the use of α-Fe(OH)_3_ nanorods with magnetic Fe and other metal oxides to induce the assembly of graphene sheets into three-dimensional graphene hydrogels used as adsorbents for water purification. The combination of electrostatic attraction, ion exchange between metal hydroxides and heavy ions, surface complexation, and oxygen-containing functional groups showed a strong adsorption capacity [41]. In addition, Wang et al. added metal ions and resorcinol-formaldehyde (RF) mixture to GO solution to generate phenolic resin as a cross-linking agent to form the gel. During the cross-linking process, metal ions interact with GO through bonding forces and promote the protonation of functional groups contained on the surface of the phenolic resin by lowering the pH of the solution to produce hydrogen bonds and thus the mechanical properties of the gel [42]. Different content (0–4 wt%) of RF will form different morphologies of 3D structures. The lower the RF content, the the more the subsequent aerogel becomes thinner, with smaller pores between the sheets and stronger electrical conductivity. Conversely, when RF content is increased, the synthetic aerogel has greater porosity and better adsorption performance [43], and the addition of cross-linker content also has an important effect on the final properties of the material.

### 3.2. Template Method for GA Fabrication

#### 3.2.1. Template-Assisted Chemical Vapor Deposition

Template-assisted chemical vapor deposition (CVD) is another widely used method to catalyze the synthesis of three-dimensional GA [9]. The GA synthesized using the CVD template method possess many advantages, such as fewer stacking interfaces between the flake layers, more complete structure, controlled and uniform pore size, and better electrical conductivity, exhibiting properties closer to those of graphene [44]. The steps of using CVD template method include the following: at the beginning, preparing a foam Ni metal substrate, then growing graphene catalytically on the substrate by CVD, and finally removing the substrate using etchant to obtain 3D graphene. The preparation of 3D graphene by CVD template method was first reported by Chen et al. Carbon generated by methane decomposition was precipitated at 1000 °C on a foam Ni template used as a 3D graphene growth, and the constant folding of graphene was shaped in three dimensions using the different thermal expansion coefficients of Ni and graphene. Then hot HCl (or FeCl) is used to etch the substrate while hot acetone is used to remove the polymethylmethacrylate that prevents the collapse of the graphene network to obtain the final three-dimensional graphene structure [9], as shown in Figure 4a. In addition to methane, Cao et al. and Chen et al. use ethanol as the carbon source, respectively, making the experimental environment safer [45,46], as shown in Figure 4b,c. What’s more, the choice of etchant is quite important. Pettes et al. used Fe(NO_3_)_3_ and (NH_4_)_2_S_2_O_8_ to remove the foam Ni, which shows gentler and less bubbles during etching. As a result, better structural integrity was obtained with better and less defects [47]. Finally, the substrate plays a considerable role as the support structure and catalyst for graphene aerogel forming. The common ones are mainly foam Cu in addition to foam Ni. Kim et al. used foam Cu to prepare graphene aerogels exhibiting good electrical conductivity, but it exhibited poor mechanical properties [48]. In this regard, Ma et al. synthesized graphene aerogel material with the special layered configuration by high temperature CVD method, which can disperse external pressure while having good resilience properties [49]. 

#### 3.2.2. Ice Template Method

Ice templates, also known as freeze casting, is another general method for the preparation of inorganic materials. It mainly refers to the solid phase (ice) generated by the phase separation that occurs during the solidification of the solution as a template [8]. As the temperature rises, the GO in the template is expelled with the sublimation of ice crystals to obtain the desired structural framework. Freeze-drying is also divided into different kinds depending on the method. Generally speaking, the faster the temperature drops, the more ice crystal nuclei are formed, which results in a uniform and small pore size of the sample. On the contrary, if the temperature drops slowly, the ice crystal nuclei near the low-temperature side form and gradually spread to other regions, generating large ice crystals and resulting in a more porous material [52]. In addition, when the reduction level of GO surface is low, the pore wall of the prepared GA is wrinkled and shrunken, and the internal GO lamellae are stacked in a disorderly manner. As the reduction level increases, the pore walls of GA are gradually flattened, and the internal GO lamellae are stacked in a tight and orderly manner [53]. Three main methods are commonly used: unidirectional freezing drying (UDF), non-directional freezing drying (NDF), and air freezing drying (AF) [26]. Firstly, ice crystals with anisotropy grow along the temperature gradient [54], and impurities can be expelled after a certain degree of freezing as the temperature changes [50]. Thus, the aerogel prepared by UFD is purer and the final formed GA has a better performance in terms of electrical and mechanical properties. Secondly, NDF does not grow according to the temperature gradient, so it is better for the overall homogeneity of the material. Finally, UFD and NDF both have smaller pore size and larger porosity for GA prepared under vacuum, so the overall adsorption capacity of UDF and NDF is higher than that of AF [55].

### 3.3. 3D Printing Methods

3D printing, as a new and unique preparation method, has obvious advantages in GA field [51], as shown in Figure 4g. It not only can the structure be designed according to the desired properties, but the homogeneity of the material can be guaranteed to a greater extent [56]. Compared to traditional GA preparation methods, 3D printing allows the design and processing of aerogels with sizes ranging from tens to hundreds of microns, addressing the key drawbacks of traditional sol-gel and mechanical processing methods. It also uses its controlled tuning to enable nano-scale GA design and precise control. This approach significantly shortens and improves the process from base chemicals or nanomaterials to useful device components, while reducing production costs [57]. This method is mainly based on the flow of GO ink material with viscosity in a certain direction under pressure and the precise deposition of a three-dimensional network structure [15]. Therefore, GO inks play a rather important role in the preparation of GA for 3D printing. Since a low concentration of GO ink does not have the rheological properties required for printing, a high concentration (10–20 mg/mL) of GO ink is beneficial to obtain high viscosity GA. In addition, adding fumed silica powder increases the viscosity of the ink, making it more adaptable for printing [51]. After supercritical drying and carbonization, the residual silica impurities can be removed [32]. Meanwhile, due to the good compatibility of 3D printing, more research on the combination of multiple preparation methods turns out to be more and more. Jiang et al. reported a cross-linking of GO solution into a viscous ink for printing by adding calcium chloride cross-linking agent [58]. Furthermore, Zhang et al. will use nozzle extrusion of GO ink to avoid drying, while using 3D printing and freeze casting methods for preparation. The advantages of both methods were combined to obtain GA with superelasticity and high electrical conductivity [59]. Although only seven years old, the development of 3D printed graphene aerogels has been effective in overcoming the limitations of current aerogel materials and has shown great potential in many applications and new market opportunities.

Synthesis of GA is an important part of GA-related research, and different synthesis methods are used to obtain materials with specified properties to meet the needs of researchers. Therefore, as the first step in GA research, synthesis occupies a fundamental and important role.

## 4. GA in the Field of Energy Storage

According to the red clusters, we can see that the application of graphene aerogel in the whole energy storage field is mainly focused on functions as an electrode material or a framework for electrode material, as evidenced by energy-storage, electrode, supercapacitors, and batteries. The main research hotspots include battery electrodes and supercapacitor electrodes [60]. For battery electrodes, although there are many other metal battery types such as sodium ion batteries, zinc ion batteries, fuel cells and so on, the most widely studied types are lithium batteries [11]. GA is useful as an electrode material not only because it has excellent electrical conductivity and stable structure, but because it is easy to compound other materials to meet the performance requirements [14].

### 4.1. Lithium Battery Electrode

Lithium battery is the most widely used battery system in the field of electrochemistry. According to the types of electrodes, lithium batteries are mainly divided into two kinds of lithium-ion batteries (LiBs) and lithium metal batteries [61], as shown in Figure 5a. Due to the good electrical conductivity and energy storage performance of the three-dimensional network porous structure of GA, they are widely used in the negative/positive electrodes of lithium-ion battery, the positive electrode of lithium-sulfur battery, and the lithium-oxygen battery. For example, 3D graphene aerogel can replace graphite anode to solve the defect of low capacity of lithium-ion battery. Meanwhile, the uniform pore distribution and large layer spacing of graphene aerogel can provide more ion channels to improve the efficiency of battery charging and discharging. In addition, the stable structure of graphene aerogel is helpful to solve the problem of service life degradation caused by volume change during the charging and discharging of battery cathode materials. The application of graphene aerogel as electrode materials has greatly improved the performance and safety of lithium batteries.

#### 4.1.1. The LiBs Anode

The anode plays the role of energy storage and release by absorbing or shedding ions during battery charging or discharging [61], and is usually a material with excellent electrical conductivity and ion storage capacity. The reactions involved in charging/discharging are as follows:Cathode: $Li⇋Li^{+}+e^{-}$Anode: $6C+XLi^{+}+Xe^{-}⇋LixC_{6}$

The most prominent anode materials for lithium-ion batteries include graphite (A mainly carbon-based material) and lithium titanate [63]. Graphene has attracted much attention as a newly discovered material in the last decades. With a honeycomb network structure, good electrical and thermal conductivity, and high chemical stability, graphene is an ideal electrode material [64], and its specific capacity (740 mAh/g) and electrical conductivity (6000 S/cm) are much higher than those of graphite electrodes with excellent energy-storage capacity. However, the battery life is limited due to the easy stacking of layers during charge/discharge cycles resulting in degraded cycling performance [65]. Therefore, using structurally stable three-dimensional graphene aerogel materials as electrodes is an effective method to solve the stacking problem.

Graphene aerogel-containing anode materials are classified into two categories based on the involvement of GA: pure graphene aerogel anodes and hybrid graphene aerogel composite anodes. Firstly, in the fabrication of pure graphene aerogel anode materials, a good structural design can enhance the electrode performance while ensuring a stable structure. In order to prevent the overlapping of layers as in the case of two-dimensional graphene, graphene oxide can be hydrothermally reduced and annealed under treated filter paper to obtain three-dimensional graphene aerogel electrodes. During the synthesis process, the filter paper acts like a rope to weave the dispersed graphene oxide flakes into a well-connected porous 3DGA. At the same time, the graphene flakes covered with folds are cross-linked with each other to form a porous structure for easy lithium ion embedding [66]. This electrode material provides a stable structure while possessing a good charge/discharge multiplicity. Besides, the choice of pore size also has a large impact on the structure. Generally, structures with microporous or mesoporous pores have a larger specific surface area and more electrochemically active reversible lithium storage sites at the electrode. The interconnection of internal active sites reduces the lithium ion embedding or de-embedding distance and accelerates the charge/discharge cycle. For example, Ren et al. fabricated hierarchical porous graphene aerogels (HPGA) with uniform and tunable mesopores (21 nm and 53 nm) on graphene nanosheets. Attributed to the mesopores capable of providing diffusion pathways for Li ions, HPGA exhibited high specific capacity and excellent cycling stability, i.e., 100 cycles with a capacity of 1100 mAh/g at 0.1 A/g current density [67]. In addition, the electrode capacity can be improved to some extent by introducing surface defects to the graphene aerogel electrode. It is shown that the increase in GA surface defects leads to the increase in active sites to a certain extent, which increases the storage sites for lithium ions and thus increases the capacity. When the defects increase to a certain number, the performance reaches a maximum value of reversible capacity 1430 mAh/g (when the current density is 0.1 A/g) [68]. To increase the GA surface defects, the first step can be taken by introducing heteroatoms. The doping of heteroatoms causes an increase in defects on the surface of graphene materials and the electrochemical activity of the material is improved due to the difference in electronegativity between heteroatoms and carbon atoms. At the same time, the electrochemical activity of the material is improved due to the difference in electronegativity between heteroatoms and carbon atoms. And the lone pair of electrons of heteroatoms as carriers can promote electron migration and ion absorption by changing the spatial structure of π-system on graphene, changing the chemical reactivity of graphene and van der Waals forces between graphene nanosheets [69]. For example, a 3D self-supporting graphene aerogel with N doping and well-controlled nanopore size for lithium-ion battery anode was reported by Yi et al. The N-doped graphene aerogel has a large number of edge sites and defects, and can provide more active sites for ion adsorption, thus enabling fast ion diffusion and charge transfer, revealing excellent energy storage capacity (1147 mAh/g at 0.1 A/g) [70]. At the same time, the dual-doped material shows stronger performance due to the synergistic effect between atoms. This S/N dual-doped graphene aerogels (SNGAs) reversible capacity reaches 1109.8 mAh/g (400 cycles at a current density of 0.8 A/g) when the content of 1.5 wt.% N and 15.11 wt.% S are co-doped in graphene aerogels [71]. Furthermore, after using high-energy photoion (3.8 MeV He) bombardment used to introduce lattice defects in the 3D interconnected network of graphene aerogels (GA), the removal or alteration of the ion beam for the functional groups on the material surface was found. This reveals the complex evolution of defects and the formation of vacancies for the enhancement of Coulomb efficiency and reversibility of graphene [69].

The use of graphene aerogels hybridized with common anode materials including metal/alloys, metal compounds, and silicon-based materials to form composites for use on lithium-ion battery electrodes is another hot research topic in recent years. These materials have good theoretical performance when used as anode alone, but suffer from various shortcomings during use. For example, these materials in ion batteries metal/alloys can charge and discharge by achieving an alloying reaction or a reverse alloying reaction, both of which theoretically have a high specific capacity [72]. However, when many charge/discharge cycles occur, the electrodes suffer from volume expansion leading to reversible capacity drop as well as safety issues. Using the structure of graphene aerogel to stabilize the nanometallic particles is a feasible method at this stage. For instance, Qin et al. designed a 3D graphene-wrapped metal Sn nanoparticles, which can not only effectively buffer the volume expansion, but provide a conductive network to promote electron transfer and improve the charge/discharge multiplicity of the battery [73]. Meanwhile, the cobalt nanoparticles produced by the decomposition of cobalt salt can react with the carbon atoms of graphene to form nanopores to constitute a multilevel porous structure. This structure facilitates ion transfer and electrochemical reactions to enhance cell performance while occupying less space through mechanical compression into thin film electrodes [74], which provides a new enhancement for the application of Li-ion batteries in portable electronics. Similar to metal/alloy electrodes, metal compound electrodes also suffer from lack of conductive network and ion transport channels, crushing or aggregation of materials caused by volume expansion, and contraction leading to batteries damage.

Metal compound electrodes, mainly consisting of two types of metal oxide (MO) electrodes and metal sulfide (MS) electrodes, utilize the Faraday reaction of metal ions to store charge and thus have the potential to achieve ultra-high energy density and large theoretical capacity (500–1000 mAh/g) [75]. It also has advantages such as low preparation cost and environmental friendliness. However, both electrodes lack conductive networks and ion transport channels, and both are prone to volume expansion and contraction-induced material crushing or aggregation during charging and discharging, leading to battery damage [76]. The conductive network of graphene aerogel and the stable 3D structure can give a good solution for the electrodes of metal compounds. GA encapsulates Fe-Co oxide to obtain Fe-Co oxide@Graphene aerogel composite, in which metal oxides are encapsulated into the GA network to prevent stacking between graphene nanosheets, and the good mechanical toughness of graphene aerogel prevents agglomeration between metal oxides. The graphene aerogel with good mechanical toughness also prevents agglomeration between the metal oxides. Importantly, when the material was compared with pure Fe-Co oxide electrodes at the same time, the former was far better than the latter in terms of electrochemical performance during use and stability of the material after multiple cycles [77]. The three-dimensional core-shell MoO_2_@few-layered MoS_2_ covered by S-doped graphene aerogel (MO/MS/SGA) prepared by Zhang et al. can not only effectively reduce the aggregation of MoO_2_, but prevent the MoS_2_ stacking. The graphene aerogel structure acts as a flexible matrix and conductive network while effectively relieving the volume buffering of Mo. The performance is good: the capacity is 683 mAh/g when cycled 100 times (current density of 0.1 A/g) [78]. The interplay between the nano-transition metal compounds and graphene aerogel in the composition of the composite is the basis for the excellent performance of the negative electrode.

Silicon is another material system with the highest theoretical specific capacity in the negative electrode of lithium-ion batteries, which is considered as one of the substitutes for carbon materials in commercial batteries [79]. However, silicon-based materials have limitations, which mainly include low electrical conductivity at room temperature, low first efficiency during charging or discharging, unstable SEI film, and poor cycling stability [80]. So far, effective progress has been made in some aspects. For example, a simple freeze-drying strategy was utilized to immobilize silicon nanoparticles on graphene flakes as anode material. The graphene aerogel porous structure acts as a mechanical support and an embedded current collector to ensure that all Si nanoparticles maintain good contact and electrochemical activity during cycling [81]. Peng et al. reported a nascent 3D SiNPs@rGO_1_/rGO_2_ composite consisting of a 3D conducting graphene backbone and a flexible pre-wrapped graphene coating as well. The special structure of graphene aerogel wrapped with silicon particles is utilized, which can improve the electrical conductivity and has a stable structural buffering volume effect. The bilayer graphene layer Is able to separate the electrolyte and silicon, thus generating a thin and stable SEI film to avoid reversible capacity loss [82].

#### 4.1.2. The Li-S Battery Electrodes

Li-S batteries are expected to replace lithium metal batteries as a new generation of efficient secondary batteries due to their high theoretical specific capacity (1675 mAh/g) and high energy density (2600 Wh/kg) [83]. Besides, this battery system embodies many advantages such as low toxicity, environmental protection, and low cost, and scientists have been enthusiastically researching it. The reactions involved in charging/discharging are as follows:Cathode: $S_{8}+16Li^{+}+16e^{-}⇋8Li_{2}S$Anode: $Li\;⇋\;Li^{+}+e^{-}$

However, the current development path has been hampered by some obstacles. The main problems include the following: (1) the poor electrical conductivity of cathode sulfur and discharge product Li_2_S; (2) the polysulfide (LiPS) produced during charging and discharging is easily dissolved in the electrolyte to form a “shuttle effect” to reduce the battery performance; (3) the large change in electrode volume during cycling, which causes damage to the structure and affects the battery life [84,85]. Graphene aerogel is a good way to solve the above problems as the positive electrode carrier structure. Firstly, graphene aerogel adopts a stable three-dimensional conductive network structure that can effectively promote the electrical conductivity of the cathode as well as prevent the volume expansion of the sulfur element during charging and discharging cycles. Secondly, the porous layered structure not only forms a channel for ion transfer, but also can be used to temporarily separate the LiPS generationed in the electrolyte and increase the sulfur loading to counteract the “dead weight” phenomenon. Thirdly, when sulfur is attached to the aerogel surface, it can provide a larger contact space for the reaction and improve the efficiency for charging and discharging [86]. For example, a GO/CNT Aerogel was proposed as a sulfur body (G/CNT-S) cathode. Thanks to the 3D interconnected porous network structure, the material presents a large surface area (336 m^2^/g) and a high electrical conductivity (67 S/m). The porosity is also able to act as a channel for ion diffusion and absorption of LiPS [87]. Meanwhile, the doping of Li-S on graphene aerogel by liquid permeation-evaporation coating as a support cathode can not only avoid the safety problems that occur during the use of lithium metal electrodes, but also the uniformly coated Li-S has a shorter ion/electron transfer pathway on the 3D graphene structure, which can effectively reduce the energy potential barrier [88].

The use of graphene aerogel-encapsulated nano MS/MO as the positive electrode material and the layered porous structure can promote ion diffusion and electron transfer, and also treats LiPS for performance enhancement by the dual effect of adsorption through van der Waals interactions and catalytic conversion of LiPS by MS/MO_2_ SnS_2_ nanoparticles embedded in graphene aerogel acting as sulfur hosts to form electrodes SnS_2_-Nano Dots@GA, which can effectively ensure a small amount of SnS_2_ in graphene aerogel with uniform deposition of SnS_2_ nanoparticles while still maintaining a high positive sulfur content (75 wt%). Using the adsorption ability of 3D graphene and SnS_2_ on LiPS, the material showed chemical stability as well as excellent performance with a capacity of up to 1016 mAh/g after 300 cycles [89]. In addition, nanosheets of other metal sulfides such as MoS_2_ with abundant defects wrapped by graphene aerogels also exhibit adsorption properties on LiPS and act as catalysts to accelerate their redox reactions [90]. Similarly, metal oxides, which are also polar nanomaterials, can provide polar-rich active sites for polysulfides as well as enhance the conductivity of electrodes. Meng et al. reported a 3D ordered ternary Fe_3_O_4_/porous carbon/graphene aerogel (Fe_3_O_4_/NC/GA). The large specific surface area and porous structure of the graphene aerogel provided the adsorption capacity for LiPS, and the highly conductive Fe_3_O_4_ embedded in the nanowafer could play a catalytic role to accelerate the conversion of LiPS [91]. Tan et al. used MnO_2_-decorated graphene aerogels in the separator for the “shuttle effect” of polysulfides, while the cathode obtained in combination with I/N-doped graphene aerogels was able to effectively immobilize and capture LiPS with long term stability—no sulfite was detected in 200 cycles substances, with intact electrode performance [92].

#### 4.1.3. The Li-O_2_ Battery Electrode

As a new electrochemical energy storage device proposed in recent years, Li-O_2_ batteries have received widespread attention for their high energy density as well as low cost [84]. Since the negative electrode material is still lithium metal, the main problem for the positive electrode of this battery is how to speed up the oxygen evolution reaction (OER) and oxygen reduction reaction (ORR) that occur during charging and discharging, respectively, the fast reaction rate will make the electrode produce large energy density and high efficiency [93]. The reactions involved in charging/discharging are as follows:Cathode: $O_{2}+2H_{2}O+4e^{-}⇋4OH^{-}$Anode: $Li⇋Li+e^{-}$

Many problems such as low cell cycle efficiency and slow reaction kinetics due to the poor conductivity of the reaction product Li_2_O_2_ need to find a suitable method to solve. The cathode of Li-oxygen battery, as the main research object, is the support for the reaction sites of the three phases (Li^+^, O_2_ and catalyst). Therefore, the following advantages are required: (1) good electrical conductivity (to facilitate electron transfer); (2) good gas diffusivity (to ensure oxygen concentration at the reaction active site); (3) porous structure, large pore capacity, and high specific surface area (to facilitate the deposition of discharge products); and (4) stable non-decomposition during charging and discharging. Based on these requirements, GA is widely used as the cathode material for lithium-oxygen batteries because of its good electrical conductivity, porosity, light weight, and relatively easy preparation. The special three-dimensional conductive network structure and high specific surface area of GA can accelerate OER and ORR, and its synthesis does not involve other superfluous substances such as adhesive, which can cause further damage to the battery performance [94], so GA has outstanding advantages as the main body of the cathode material. This is mainly reflected in the high specific surface area that enables more active sites, while the high number of pores facilitates electrolyte and oxygen transport. The excellent electrical conductivity of the network structure allows a faster rate of electron transfer as well as facilitating the kinetics of the reaction process [95,96]. At the same time, the porous structure can be used to accommodate Li_2_O_2_ to reduce deposition and increase the capacity of Li-O_x_ batteries. For example, Zhao et al. used carbon nanocages prepared from polystyrene sphere@polydopamine (PS@PDA) and then hydrothermally self-assembled coupled into a 3D porous N-doped graphene aerogel (NPGA). The material has a crumpled and loose structure that prevents the stacking of graphene sheets thereby exposing more reactive sites. Compared to the GA without the carbon nanocage structure, NPGA acts similarly as a catalyst, increasing the ORR onset potential and decreasing the OER onset potential with a faster reaction rate [97]. Not only that, but the team also prepared a stable carbon nanofiber graphene aerogel film (G-CNF film) as the main material for the electrode of Li-O_x_ battery. As a two-in-one material for positive and negative electrodes, it can not only reduce the formation of negative Li dendrites to increase the battery life, but mitigate the bulk effect to increase the battery capacity and ensure the cycle efficiency [95]. Therefore, in the research related to the application of GA in battery electrode materials, whether as the main framework or composite with other materials, its good conductivity or stable three-dimensional support structure plays a rather critical role.

### 4.2. Supercapacitor Electrodes

Supercapacitor, also known as Ultracapacitor, is an energy storage device with a large capacity [98]. According to the energy storage mechanism, they are mainly divided into two types: electrochemical double layer capacitors (EDLC), which use the absorption and desorption of ions on the electrode surface to form a double layer of charge, and pseudocapacitors, which generate charge through a Faraday process on the electrode surface [24,99,100], as shown in Figure 5b. As an important dependency of a supercapacitor device, the electrode material is crucial to its performance. The specific surface area and porosity of electrode materials are two important factors to measure the performance of double layer capacitors, but few carbon materials can satisfy both. In particular, redox pseudocapacitors have attracted the attention of researchers due to their ability to provide higher capacity. However, the materials are limited by their own properties, such as poor electrical conductivity, and the long redox reactions during the charging and discharging process tend to cause damage to the material structure, resulting in lower energy density and cycling stability performance [101]. As 3D stable porous carbon materials, graphene aerogels have high electrical conductivity and strong mechanical strength, which provides a fast and convenient channel for the storage and transfer of electrons in EDLC, improving capacitance and efficiency [102]. At the same time, graphene aerogels with high specific surface area and nano-effects in pseudocapacitors can also be effective for traditional materials such as metal oxides or hydroxides with low energy density and cyclic stability [16].

#### 4.2.1. Electrochemical Double Layer Capacitor Electrodes

The main reason for that graphene aerogels can be used as electrode materials in double-layer capacitors is due to their good electrical conductivity and structural properties [103]. When the electrode is immersed in the electrolyte, the structure has a catalytic structure of mesopores and micropores that facilitates the formation of EDLC double layer capacitors [104]. At the same time, the surface macropores are filled while acting as buffer channels for ion diffusion, shortening the ion diffusion time to facilitate ion storage and transfer in the electrolyte [105]. As a representative example, Zhu et al. used direct-ink writing (DIW) to prepare a compressible graphene aerogel network lattice. The GO ink printed graphene aerogel electrodes doped with optimised graphene nanoplatelets have almost constant large specific surface area and lower resistance compared to normal 3D printing to prepare graphene aerogels. The orderly arrangement of the large pore structure promotes the transfer of material and reduces the diffusion resistance in the structure, thus ensuring that the electrode has excellent rate capability and electrochemical stability [106]. At this point, the GA Liquid Crystals (LCs) prepared by Xu et al. have a unique “porous core-dense shell” structure that gives the material good tensile and compressive strength, with a maximum compressive modulus of 3.3 Mpa and can withstand more than 2000 times its own weight without damage. The conductivity is also considerably higher than that of ordinary GA (4.9 × 10^3^ S/m) and remains stable after 100 bending cycles [60]. While in practical application, microporous channels may reduce the rate of ion diffusion in the electrolyte at high loading rates, and the longer the pipe, the slower the diffusion [107]. Therefore, when designing graphene aerogel electrodes, attention should be paid to the length of the pore channels. In addition, Fan et al. reported a unique sheet-like vortex-connected structure of graphene-carbon nanotube aerogel (GCA) as a conducting framework for high theoretical capacity nickel hydroxide. This material improves the performance of the electrode by maximising the exposure of active sites through a convoluted interconnected nanostructure that hinders the stacking of graphene sheets and the generation of mesopores [108].

Graphene aerogel-related composites also offer superior performance when used as electrochemical double-layer capacitor electrode materials. Typically, the electrical conductivity and mechanical stability of metal oxides can be improved when combined with 3D graphene [109]. Tian et al. synthesized a composite aerogel with a three-dimensional layered porous structure by immobilizing MnO_2_ on RGO, which was also effective in controlling graphene sheet agglomeration and expanding the electrode capacitance. At a current density of 1 A/g, it exhibited a high specific capacitance of 645 F/g [110]. However, due to its insufficient capacitance retention, Chen et al. further used NiO to replace MnO_2_ for the composite with GA. In the course of performing cyclic stability tests, it was found that the specific capacitance of this NiO/GA material remained almost constant and had excellent stability [109]. Composites typically have higher electrical capacity and electrochemical stability than pure GA, which is very important for EDLC, enhancing the efficiency and lifetime of supercapacitors

#### 4.2.2. Pseudocapacitor Electrodes

Compared to EDLC, pseudocapacitor electrode materials transfer electrons and store energy through redox reactions between the electrode surface and the electrolyte, resulting in greater adsorption and thus greater capacitance [111]. Since the reactivity of the electrode surface has a large impact on the pseudocapacitor, it is a good way to improve the performance by improving the active sites on the electrode surface [112]. Three-dimensional graphene has a large specific surface area and can be doped with eroatoms (N, S, B, etc.) to be able to present more active sites for surface reactions and more effectively improve the wettability of electrodes and electrolytes to promote electron transfer and thus increase the capacitance of supercapacitors. For example, 3D nitrogen-doped GA nanomesh (N-GANM) can exhibit a high specific capacitance of 290.03 F/g at a current density of 1 A/g, which is much higher than the 197.5 F/g of ordinary graphene aerogel [113]. Besides, the nitrogen atoms weakened the stacking of graphene sheets and increased the specific surface area of the electrode material, which was able to accelerate the reaction rate with the electrolyte [114]. Compared to single doping, graphene aerogels used as electrodes with multiple doped atoms exhibit better results [115]. Double nitrogen-doped materials possess more nitrogen atoms and are more likely to increase the subversion of the reaction sites to accelerate ion diffusion [116]. Double doping of N and S can improve the electroneutrality of graphene and facilitate the rapid diffusion of ions from the electrolyte into the electrode material to increase the charge/discharge rate and improve the cycling stability [117]. The double doping of N and B can effectively promote the formation of B-N motifs in the carbon matrix, generating local surface polarization and thus increasing the redox sites [118]. In addition to this, the doping of heteroatoms leads to the creation of surface defects and the appearance of more vacancies. The appearance of these vacancies disrupts the original left-right symmetry of the system and allows the electronic structure near the Fermi energy level to change, thus enhancing the conductivity [119]. Generally, metal compounds have high theoretical specific capacitance and wide availability, so recently GA and metal compound nanoparticle composites have also been used as pseudocapacitive electrode materials. Yao et al. used GA as a scaffold and 3D printed graphene/MnO_2_ compounded with MnO_2_ on its surface using 3D printing technology as an electrode material. It was found that, to a certain extent, the electric capacity increased with increasing MnO_2_ loading content and the weight of the electrode material remained almost constant. This shows the ability to substantially increase the loading of the pseudocapacitor material without sacrificing rate capability and weight capacitance, which is essential for the development of practically useful pseudocapacitors [120]. The inherent well-defined porosity and high surface area of metal-organic frameworks (MOFs) and the three-dimensional structure after compounding with GA allow electrolyte ions to diffuse in and out of the network even in thick electrodes with high quality loading, which is the outstanding advantage. Barua et al. generated pseudocapacitance by compounding Co-containing MOF with GA using redox-active ligands composed of hydroquinone units. Strong chemical interfacial engineering between MOF and graphene enhances the ion adsorption energy during electrochemistry and contributes to the charge storage performance [121]. The development of GA in supercapacitors has greatly contributed to the progress in the selection of electrode materials, while realizing a high electric capacity that is beyond the reach of many materials.

The emergence of GA has brought a big change to the energy storage, making battery systems and supercapacitors show high energy conversion efficiency and good development prospects. With its lighter material and larger energy storage capacity, it is expected to replace the traditional carbon material electrode as the new favorite of scientific researchers. However, the practical application of the whole GA material has not been well solved, and a perfect system to promote the practice is needed. There is still a long way to go to realize the application in energy storage field.

According to the keywords presented in the blue clusters, the main research hotspot are Adsorption and Removal, which clearly shows another major application area of GA: environmental protection. Since the 3D porous structure has good adsorption and absorption ability as well as large specific surface area for catalyst loading, GA plays a great role in water pollution and air pollution control methods.

## 5. Application of GA in the Environmental Protection Field

### 5.1. Adsorption

The porous structure and large specific surface area of aerogels determine their excellent adsorption of pollutants [18]. Moreover, with exceptional resilience it is easy to separate the adsorbed material for recycling and utilization [122]. Therefore, GA is expected to be an efficient adsorbent to absorb pollutants in water and gases.

#### 5.1.1. Adsorption of Wastewater Pollutants

In recent years, the number of offshore oil spills has been increasing and the leaked crude oil has seriously affected the balance of the surrounding ecosystem [123]. The effect of capillary driving force in the nanochannels of graphene materials leads to low diffusion rate of water molecules between graphene sheets with hydrophobic stability [124]. With the combination of oxygen-containing functional groups on the GA surface structure that can interact with hydrophilic substances and arylalkyl groups that interact with hydrophobic substances, they have a strong adsorption capacity for organic substances such as oil with the help of Π-Π interactions and electrostatic gravitational forces [123,125,126]. Therefore, GA is widely used to absorb organic waste from water to purify water.

In 2016, Liu et al. prepared anisotropic graphene aerogels (AGAs) with lipophilic hydrophobicity and efficient adsorption properties using unidirectional freeze casting. The oxygen-containing functional groups on the surface of GA are quite compatible with many organic polymers and have a good adsorption effect on n-heptane, ethanol, acetone, etc. It can adsorb organic masses more than 120–200 times its own and can be easily recovered by combustion, distillation, and extrusion [125] Figure 6a–c. Meanwhile, graphene aerogel-based composites can effectively regulate the pro/hydrophobic properties as adsorption by adding other substances, and have some other environmental protection concepts. For example, Hu et al. prepared a multifunctional composite aerogel using chitosan (CS) as the backbone substrate and reduced graphene oxide nanosheets as the reinforcing agent. The material not only maintains good adsorption capacity and stability in harsh environments but can be easily degraded [127]. Moreover, Sun et al. synthesized a 3D macrostructure graphene-CuFeSe_2_ aerogel for crude oil adsorption using solar energy drive. The photoexcitation heat generated by the photothermal material CuFeSe_2_ to heat the crude oil drives its diffusion into the GA, and oil can be retained in the pores of aerogel after the removal of the light source. The method low cost and in line with the green development concept [128].

Pollution caused by heavy metal ions leaking into water bodies when produced by industrial manufacturing can also seriously endanger human health and damage the ecosystem in water [126]. GA with electronegative functional groups on the surface are able to combine with different metal ions, and thus used for metal waste purification. When using GA for adsorption treatment, the adsorption capacity of GA is influenced by various factors (Ph, temperature, surface functional groups, etc.), so needs to be considered. For example, under acidic conditions, the increase in ionization of functional groups on the GA surface leads to an increase in the amount of negative charge, when electrostatic gravity plays an important role in the adsorption of metal ions. When the pH value in the solution gradually increases, the functional groups on the GA surface combine with metal ions to form complexes, and complexation and metal ion exchange become the main mechanism of GA adsorption [130]. Meanwhile, the adsorption capacity of GA increased with the increase in temperature. Since the adsorption reaction requires heat absorption, the temperature decreases during the reaction and the adsorption capacity of GA gradually decreases. In addition, the presence of functional groups helps to expose the surface-active sites of GA, giving it a higher adsorption capacity [131]. A tetraethylenetetramine/polypyrrole/graphene oxide aerogel (TPGA) was prepared by Liang et al. Under acidic conditions, the amino functional group promoted the adsorption of Cr^2+^ up to 408 mg/g [33]. Pan et al. reported that a novel macroporous calcium alginate/graphene oxide composite aerogel (mp-CA/GO) was able to effectively adsorb Pb^2+^, Cu^2+^, and Cr^2+^ while achieving remarkable removal rates of 95.4%, 81.2%, and 73.2%, respectively [132]. It can be noted that, based on the strong conductivity of GA, electroabsorption, which removes metal ions from adsorbed water by forming an electrostatic field on the surface of GA electrodes through an applied voltage, is also an efficient way of water purification. Wei et al. reported the use of nitrogen-doped graphene aerogel (NGA) for the removal of lead ions from wastewater by electrosorption. NGA was added to the Pb electrode to improve its mechanical strength and adsorption capacity. It also remained stable at 100 times of adsorption-desorption [133].

#### 5.1.2. Adsorption of Gases

The current applications of GA in gas adsorption include the adsorption of toxic gases (CO, H_2_S, NO, etc.) or greenhouse gases (mainly CO_2_) and the adsorption-desorption reuse of energy gases (mainly H_2_, CH_4_).

Leakage of toxic gases is detrimental to both human and environment, so adsorption and subsequent treatment of leaked gases is important for environmental protection [134]. Due to the large degree of environmental hazards, the treatment of toxic inorganic gases such as H_2_S and CO by GA includes subsequent auxiliary catalytic conversion to non-toxic substances in addition to adsorption. For example, three-dimensional flexible alkaline graphene aerogel (AGA) was prepared using Na_2_CO_3_ and GA to adsorb and catalyze the oxidation of H_2_S. The macroporous structure of AGA provides large capacity for adsorption of H_2_S and the alkaline environment provided by the Na_2_CO_3_ attached on the large specific surface area promotes the dissociation of H^+^ from H_2_S. Oxygen molecules are activated as superoxide radicals on the surface of graphene sheets, which promotes the catalytic oxidation of HS^-^ to sulfur monomers [135]. In addition, Qu et al. took advantage of the large specific surface area and strong adsorption capacity of MOF materials to adsorb CO by synthesizing 3D Ru/GA-HK with HKUST-1 (HK) modified on the GA surface. The macroporous structure of GA provides a pathway for the transport of gas molecules, allowing CO to diffuse between the graphene and Ru layers to form CO_2_ using oxygen oxidation. HK loading on the GA surface enhanced the adsorption capacity of CO and increased the catalytic efficiency by ≈48.4% [129]. Apart from toxic gases, excessive CO_2_ emissions are the main cause of the intensification of the greenhouse effect, and reasonable control of CO_2_ content is of great importance [136,137]. Xia et al. compounded MgAl-LDH/RGO aerogel by layer double hydroxide (LDH) with MgAl nanoparticles added to GO, which was used to study the effect on CO_2_ capture at high temperature and pressure. Compared to the two-dimensional MgAl-LDH powder, the adsorption efficiency of the three-dimensional porous structure of MgAl-LDH/RGO aerogel was improved by about 160% [138]. Since basic amine groups can react with acidic CO_2_ gas by condensation, adsorbents containing amine groups can be used to adsorb CO_2_. Therefore, CS-embedded GO aerogels were used for CO_2_ adsorption by Hsan et al. The large number of amine groups contained inside the CS attached to the GA surface can have a larger reaction area to react with CO_2_, much higher than the adsorption content of pure CS [139]. Leakage of energy gases causes waste of resources, and the treatment of energy gases such as H_2_ should ensure the efficiency and stability of adsorption-desorption. Peng et al. reported the shell-core MgH_2_@carbon aerogel microspheres for the storage of H_2_. Thanks to the bonding between Mg and hydrogen atoms and the microsphere structure that inhibits the agglomeration of the material, the material exhibits a very good adsorption capacity and high cycling stability to meet the requirements of long-term use [140]. After adsorption, the desorption process can be carried out by physical extrusion using the resilience of GA or by creating an acidic environment to increase the number of H ions to replace the heavy metal ions in the active site [141].

### 5.2. Removal

In addition to adsorption, GA photocatalytic carriers are also a good method for catalytic removal of pollutants. 3D GA has a macroscopic ontological structure and is an ideal carrier for photocatalysts [129]. It can provide a stable backbone structure and large contact area for different photocatalysts. The targets of the scavenging process include a few organic compounds and microorganisms.

Generally, organic compounds are removed by degradation to water and CO_2_ with GA-loaded surface catalysts [142]. For instance, a 3D TiO_2_/reduced graphene oxide aerogel composite was prepared by Qiu et al. The catalyst TiO_2_ grows along the glucose on the surface of GA and interacts to form a stable structure, while the large specific surface area provided by GA accelerates the catalytic degradation of methyl orange (MO) by TiO_2_ (up to 90% degradation rate by sunlight irradiation for 300 min). At the end of degradation, GA and TiO_2_ can be easily separated by physical manipulation, which is very convenient [143]. At the same time, multiple complexes composed of multiple functional compounds coupled together have higher catalytic properties. BiVO_4_/RGO aerogel/CeVO_4_ (BGC) degradation of tetracycline (TC) in visible light was proposed by Liu et al. The high electron mobility inside the material under the joint action of the three compounds enables a more efficient separation of photogenerated electric carriers. And compared to pure BiVO_4_ and CeVO_4_, the degradation rate was increased by 66% and 69% using GA as the backbone structural material [144]. In addition, certain heteroatom doping also helps to improve the performance of photocatalysts. Boron-doped graphene aerogels with different concentrations were prepared by Chowdhury et al. for the catalytic degradation of acridine orange (AO). The doping of B opens the band gap of graphene and increases the activity efficiency of the catalyst by ≈2 times (compared to the catalyst without GA incorporation) [145]. Additionally, microorganisms (bacteria, fungi, viruses, etc.) in wastewater also have a harmful effect on human health and the environment, and their removal using photocatalysis is also a current research hotspot.

Microbial scavenging mainly uses the products (ions, electrons, free radicals, etc.) obtained from redox reactions induced by active substances to cause damage to cell structures to achieve inactivation, and GA as a carrier template provides a large amount of reaction space and contact area with microorganisms [146]. For example, Zhang et al. used the prepared Ag-AgBr/TiO_2_/GA to perform degradation inactivation studies on *E. coli* and *S. aureus*. AgBr nanoparticles are effectively combined with TiO_2_ nanocrystals to accelerate the light-induced charge transfer and separation to achieve the decomposition of *E. coli* and *S. aureus*. The backbone structure was provided by GA and the large specific surface area promotes Ag-AgBr light collection and reaction platform [147]. In addition, three-dimensional BiOCl/RGO aerogels prepared by Zhang et al. were used for the catalytic decomposition of oxytetracycline (OTC). The porous structure of three-dimensional graphene with high specific surface area has abundant active sites providing powerful adsorption capacity of BGA composites. Besides, the material has enhanced visible light response and effective separation of electrons and holes, which are important for the antibacterial activity of OTC [148].

In environmental protection, the porous structure and surface functional groups of graphene play an important role. Meanwhile, more and more three-dimensional composites based on graphene aerogels are emerging, demonstrating the performance advantages and wide practicality of graphene aerogels.

## 6. Conclusions and Outlook

The development of nanotechnology has greatly contributed to the interest in various fields. Surprisingly, the related three-dimensional material GA has made considerable achievements in the past decade, and its three-dimensional stable structure has shown superiority over two-dimensional graphene in many aspects. The review uses bibliometric methods to analyze three hot areas of GA fabrication methods, energy storage applications, and environmental applications.
(1)We firstly discuss the self-assembly method, the template method, and the 3D printing preparation of GA, respectively, and analyze the advantages and limitations of different preparation methods. We found that the assembly method as a commonly used preparation method has the advantages of easy operation and low cost, and is expected to be produced in large quantities on a factory scale. However, the molded GA surface often contains many defects and collapses, which can seriously impair performance. The GA obtained by template method and 3D printing has better results in terms of material structure and uniformity of properties as well as stability in long-term use. However, the relatively high cost of these two methods and the tedious preparation process makes them difficult to meet in industrial scale manufacturing at present.(2)GA with its excellent conductivity and large specific surface area is used as an electrode for batteries and supercapacitors to store energy. In lithium battery systems, GA is usually used as an electrode material alone or in combination with other electrode materials. Its good electrical conductivity and large specific surface area can effectively improve the transfer efficiency of lithium ions during charging and discharging to achieve rapid charging and discharging. In supercapacitors, the porous structure provided by GA can increase the electrode capacity as well as provide electrochemically active sites for electron attachment after doping with heteroatoms to achieve high energy density. The large specific surface area and high electrical conductivity possessed by GA are its advantages, and it has a huge advantage in the development of energy storage.(3)In the field of environmental protection, the hydrophobic properties of the oxygen-containing functional groups on the surface of GA combined with the porous structure enable effective adsorption of organic substances, and its good resilience can effectively achieve multiple cycles of adsorption-desorption under the action of mechanical pressure. Meanwhile the large surface area provides loading for photocatalysts to improve the efficiency of catalytic degradation of pollutants, which will play a very significant role in promoting the improvement of environmental problems. The development and progress of GA is closely related to the development of these three major research fields. As a visualization method, the bibliometric analysis presents objective and comprehensive content. The summary of the key sections will be presented to the reader more clearly and quickly.

Currently, more and more studies on GA are being reported in a wider range of applications, for example, as electromagnetic shielding materials to reduce electromagnetic interference and as biosensors for sensitive detection of cells [149,150,151,152,153]. However, there are some challenges in the development of GA. The preparation of controlled structures with ordered arrangements and the loss of performance during use still pose limitations to its development, and the fundamental reason is the lack of most of the research between the transition from 2D graphene to 3D GA. Therefore, a more detailed study of the mechanism from both theoretical and experimental aspects is necessary. Challenges come along with opportunities, and it is foreseeable that GA will have more room for performance in the future.

## Figures and Tables

**Figure 1 materials-16-00272-f001:**
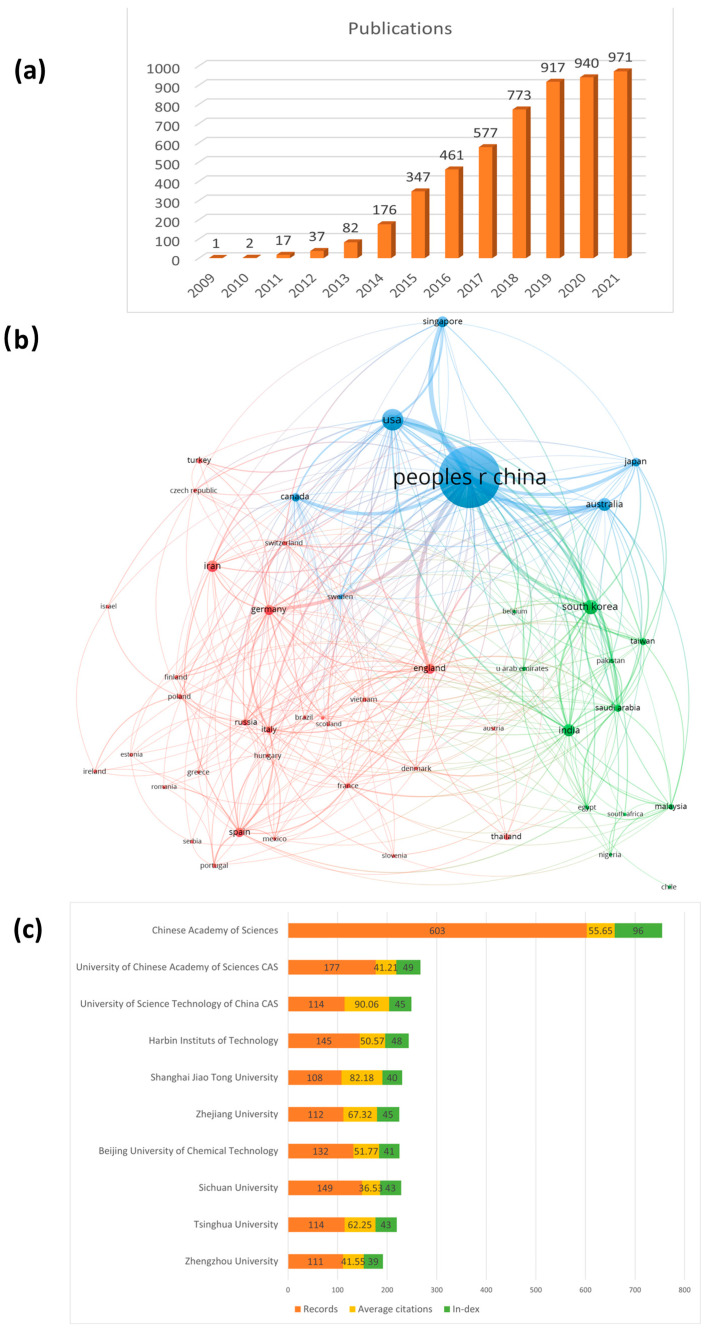
(**a**) Annual number of publications related to GA from 2009 to 2021. (**b**) VOSviewer shows the cooperation between different countries/regions on GA research. (**c**) The most productive institutions based on the total publication.

**Figure 2 materials-16-00272-f002:**
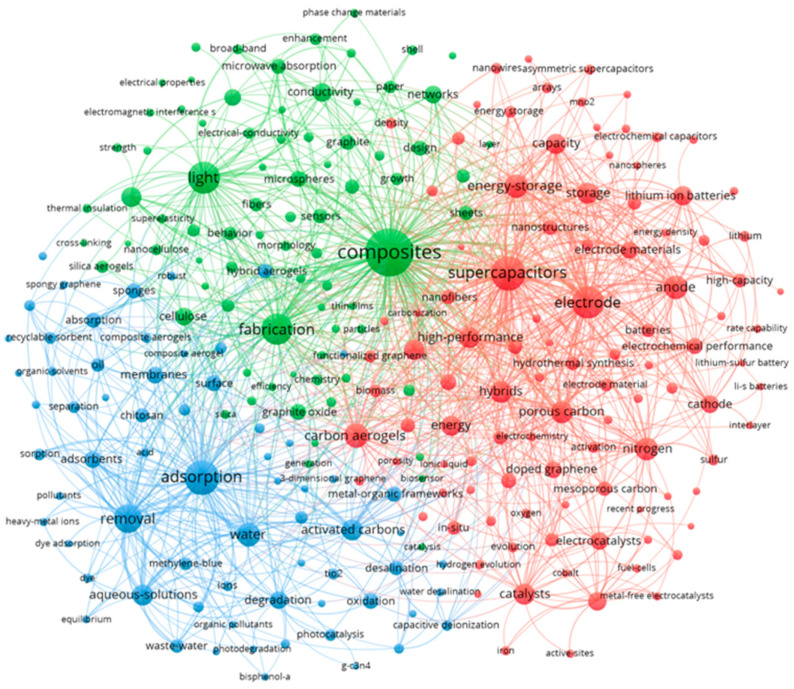
Summary network diagram of GA’s bibliometric analysis. Keyword co-occurrence analysis with at least 20 occurrences. The node size represents the occurrence frequency of the keywords. The same color represents relevance.

**Figure 4 materials-16-00272-f004:**
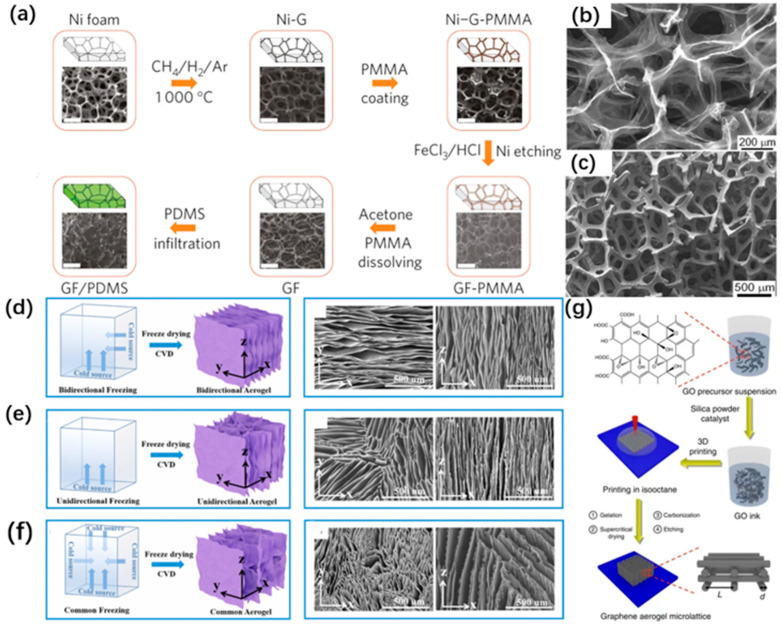
(**a**) CVD growth of 3D graphene using nickel foam as a template [9]; (**b**). SEM image of 3D graphene network grown on Ni foam after CVD; (**c**) SEM image of 3D graphene network after removing Ni foam [46]; (**d**) Schematic diagram and SEM image of GA prepared by two-way freezing casting; (**e**) Schematic diagram and SEM image of GA prepared by unidirectional freezing casting; (**f**) Schematic diagram and SEM image of GA prepared by freezing casting and common freezing casting [50]; (**g**) Schematic diagram of 3D printing GA process [51].

**Figure 5 materials-16-00272-f005:**
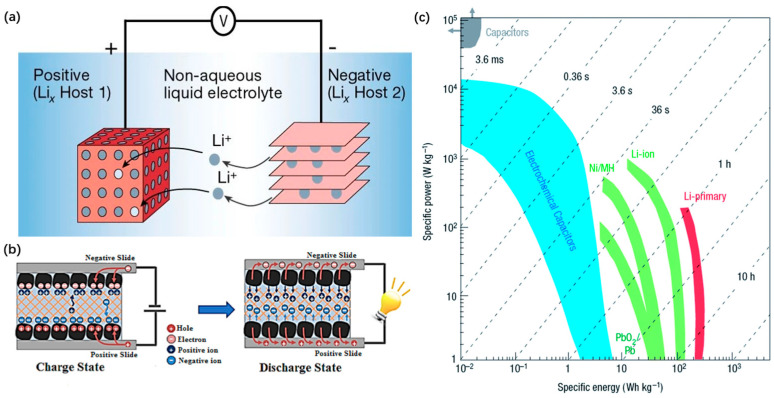
(**a**) Schematic diagram of a rechargeable lithium-ion battery [61]; (**b**) Charging and discharging states of supercapacitors [24]; (**c**) Specific power density against specific energy density, also called a Ragone plot, for various electrical energy storage devices [62].

**Figure 6 materials-16-00272-f006:**
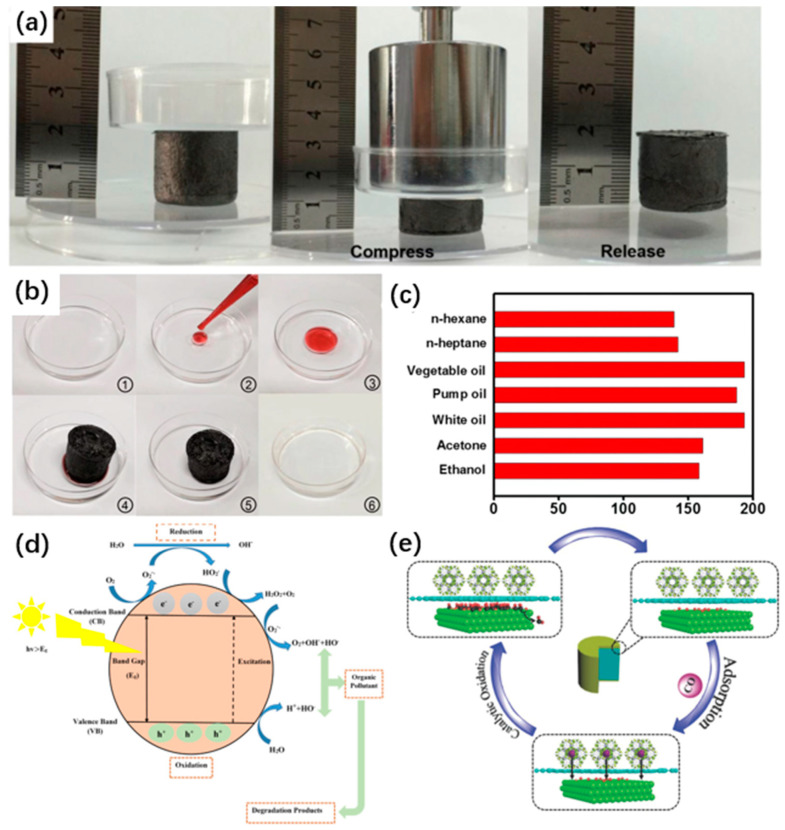
(**a**) Digital images show that AGA-8.3 (height = 18.6 mm, diameter = 21.3 mm, mass = 55 mg) can support 500 g weight and exhibits a good compressive performance; (**b**) Digital images showing the removal of n-heptane that is stained with Sudan Red 5B and floating on water [125]; (**c**) The ability of AGA-4.2 to absorb different organic liquids; (**d**) Photoactivity degradation mechanism of organic pollutants [19]; (**e**) Schematic of the reaction mechanism for CO removal [129].

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
