# Peer review of "Progress in Research and Application of Graphene Aerogel—A Bibliometric Analysis"

_materials, 2022, doi:10.3390/ma16010272_

Round 1

Reviewer 1 Report

Reviewers' comments:

Manuscript ID: materials-2073872

Title: Progress in research and application of graphene aerogel -- based on bibliometric analysis.

Manuscript Type: Review

Reviewers' comments:

The manuscript describes the Progress in research and application of graphene aerogel -- based on bibliometric analysis. The manuscript needs a detailed editing. Some markings are made to just illustrate the extent of editing needed. A thorough revision addressing all the concerns is needed and if the authors are prepared to do that it can be considered for a review of the revised manuscript.

The authors need to consider the following comments

- In the Abstract the authors need to improve with more specific short results.

- Keywords: author should add more suitable keywords.

- The introduction is poor and less informative. Authors should elaborate their introduction section by citing few more relevant references. The novelty of the work should also be highlighted.

- The quality of figures 1 is too low.

- Figure 2 - not clear make clear.

- 3.2.2. Ice template method - should be improve.

- 3.3. Other novel methods for GA fabrication - should be improve.

- 4.1.3 The Li-O2 battery electrode - should be improve.

- Line number 420 - The Li-O2 battery electrode……to…….. The Li-O2 battery electrode

- Line number 498 - reaction product Li2O2 need…to...reaction product Li2O2 need

- Line number 507 - accommodate Li2O2….to…accommodate Li2O2

- 4.2. Supercapacitor electrodes - should be improve.

- Line numbers 569 and 572 - MnO2…to…MnO2

- The author should use proper chemical formulas for the whole manuscript (for example – CO2…to…CO2, H2…to…H2, MgH2….to….MgH2, Co3+….to…Co3+, and etc….).

- The Conclusion and Outlook part needs to be reorganized; innovation points need to be further clarified.

- References: author should use order and there are recent references in 2021-2022 treating the same subject, you can use.

- Make all references in same format for volume number, page number and journal name, because it is difficult to searching and reading.

- Language needs substantial improvement. Please consult a native English speaker or a language editing service.

Based on these, I advise the authors to rectify the above mentioned errors and we hope to re-evaluate the revised manuscript.

Author Response

  1. In the Abstract the authors need to improve with more specific short results.
  2. Keywords: author should add more suitable keywords.
  3. The introduction is poor and less informative. Authors should elaborate their introduction section by citing few more relevant references. The novelty of the work should also be highlighted.

Response: Thanks for your suggestion. We have added major results to Abstract, suitable words to Keywords and provided more information in Introduction part. The novelty is also highlighted at the end paragraph of Introduction.

  1. The quality of figures 1 is too low.
  2. Figure 2 - not clear make clear.

Response: Thanks for your suggestion. We put the fourth and fifth questions in one response. We have replaced the old images with high resolution image of figure 1 and 2.

  1. 3.2.2 Ice template method - should be improve.

Response:Thank you for your comments. We have now refined the ice template method in the manuscript (page10/line number273-281).

  1. 3.3 Other novel methods for GA fabrication - should be improve.

Response: Thank you for your comments. Since the scope of our review is to identify the major hotspot of research related to GA, we thus only provide most widely studied research topic. We are sorry that we used title of "Other novel methods" in our original manuscript, which doesn’t highlight the research hotspot. We thus changed the title to "3D printing method" and improved the content (page11/line number295-300).

  1. 4.1.3 The Li-O2 battery electrode - should be improve. (Line number 420 - The Li-O2 battery electrode……to…….. The Li-O2 battery electrode, Line number 498 - reaction product Li2O2 need…to...reaction product Li2O2 need, Line number 507 - accommodate Li2O2….to…accommodate Li2O2).

Response: Thank you for your comments. We have improved the content of lithium-oxygen batteries in (page16/line number541-550), hope you will find it acceptable.

  1. 4.2 Supercapacitor electrodes - should be improve. (Line numbers 569 and 572 - MnO2…to…MnO2).

Response: Thank you for your comments. The supercapacitor part is a major application of graphene energy storage, and we have written about it by splitting it into two aspects, EDLC and pseudocapacitor. We have refined and added explanations to the relevant contents (page16/line number581-588, 660-676).

  1. The author should use proper chemical formulas for the whole manuscript (for example – CO2…to…CO2, H2…to…H2, MgH2….to….MgH2, Co3+….to…Co3+, and etc….).

Response: Thank you for your comments. We have used proper chemical formulas for the whole manuscript.

  1. The Conclusion and Outlook part needs to be reorganized; innovation points need to be further clarified.

Response: Thank you for your valuable advice. We have corrected the conclusion section and clarified the Outlook part using bullet points.

  1. References: author should use order and there are recent references in 2021-2022 treating the same subject, you can use.
  2. Make all references in same format for volume number, page number and journal name, because it is difficult to searching and reading.

Response:Thank you for your advice and reminder. We have used some relevant literature and content published recently in the revised manuscript including:

DOI10.1016/j.cej.2022.139793

DOI10.1016/j.cej.2022.139338

DOI10.1016/j.scriptamat.2017.06.020

DOI10.1021/acs.nanolett.0c04780

DOI10.1007/s10118-022-2707-3

DOI10.1039/c9cs00757a

DOI10.3390/nano11051240

DOI10.1016/j.joule.2018.09.020

DOI10.1016/j.jclepro.2020.125776

DOI10.1021/acsaem.1c02991

DOI10.1016/j.cej.2022.139376

DOI10.1021/acsnano.0c09982

DOI10.1021/acs.analchem.1c05293

In addition, we have sorted out the volume number, page number and journal name of the reference section to ensure that it will not be difficult to search and read.

  1. Language needs substantial improvement. Please consult a native English speaker or a language editing service.

Response:Thank you very much for your kind suggestion. We have tried our best to correcting English language. Due to the time limits, later we will use MDPI language service for further assistance.

Reviewer 2 Report

Comments: materials-2073872

In the present investigations, the authors examine the experimental investigation of Progress in research and application of graphene aerogel -- a bibliometric analysis. The topic of study is fascinating and well-developed. I advise publication the manuscript after some minor changes.

·         1. Please add the novelty of the problem prior to the previously published work.

·   2. The abstract should be enhanced with some major results. If possible add the qualitative results.

·        3. Why do authors consider Graphene Aerogel? What is the advantage?

·        4. What are the advantages of the bibliometric method compared to other methods?

Author Response

  1. Please add the novelty of the problem prior to the previously published work.

Response: Due to the wide range of GA applications, reviews that summarizes its application have been widely reported. However, there still lacks of a work that can provide an overview of the popular research areas from a holistic and objective perspective, which could help researchers to better understand and grasp the key application directions related to GA. Therefore, an objective, timely and comprehensive statistical approach to GA-related research is needed. Our work draws on bibliometric visual analysis as a method to develop a review of GA-related studies. Unlike other articles, the most prominent feature of this paper is the presentation of the objectively identified hot areas of GA, which facilitates the reader to understand more clearly the mainstream direction of current research. In view of this, we for the first time use the bibliometric analysis method to conduct a comprehensive and timely review regarding GA, to identify current research states and promote the growth of this vital topic. Bibliometrics refers to the quantitative analysis of literature information using mathematical and statistical methods, and is often used to determine general knowledge frameworks, assess current state, and predict future directions in a given field. It is anticipated that such a review article can help readers to grasp a general research status as well as give important insights for guiding the future development of GA research. Hope you are satisfied with our response.

  1. The abstract should be enhanced with some major results. If possible add the qualitative results.

Response: Thank for you suggestion. We have added major results to abstract.

  1. Why do authors consider Graphene Aerogel? What is the advantage?

Response: Graphene aerogel is essentially a three-dimensional graphene material with aerogel structure, and both graphene and aerogel are one of the popular functional materials in recent years. GA possesses all the characteristics of graphene and aerogel at the same time, with more outstanding performance. Therefore, it has been widely used in energy storage and adsorption, which can’t be realized by graphene and other aerogel such as silica aerogel. GA has excellent properties such as low density, high electrical conductivity, high thermal insulation, large specific surface area, good resilience and strong adsorption ability, which can effectively be used in important filed including resource shortage and environmental pollution. Its high electrical conductivity and stable three-dimensional structure as electrodes in electrochemical energy storage devices for energy applications, its porous structure and resilience properties enable their application as an adsorbent and solar photocatalytic carrier in environmental remediation, and its compressibility and sensitivity to gas and liquid for use as sensors, etc. The emergence of graphene aero-gels has greatly facilitated the development and application of graphene-related materials, which can be effectively used in important situations such as resource shortage and environmental pollution.

  1. What are the advantages of the bibliometric method compared to other methods?

Response: Since there is a large amount of literature related to GA, the ordinary way of information collection requires a meticulous review of a huge amount of literature and is difficult to cover all the information from a holistic perspective. Bibliometrics is a statistical method that can analyze a large number of publications, to qualitatively and quantitatively evaluate the research performance of countries, institutions, and certain subject domain. Alternatively, bibliometric analysis can also objectively analyze the current situation and indicate the future development of a specific field. Therefore, in this article, upon identification of all the publications regarding GA, we employed bibliometric method to study the dataset by year, country, institute, as well as research hotspots. Based on the objectively identified research hotspots from VOSviewer, we further provided a detailed overview on GA fabrication, energy storage and environmental protection. In another word, compared with traditional reviews, the structure of our review is based on the comprehensive and objective analysis of the research field of interest using a bibliometric method, which provides a holistic angle to analyze the research performance of this field. What’s more, bibliometric analysis method could also combine visualization method to clearly and directly grasp the major research trends in this field. Hope we have explained the advantage of bibliometric analysis well.

Round 2

Reviewer 1 Report

The manuscript can published. The authors have answered the questions.